# Assessing the influence of culture on craft skills: A quantitative study with expert Nepalese potters

Enora Gandon[1]*, Tetsushi Nonaka[2], Raphael Sonabend[3], John Endler[4]

1 Institute of Archaeology, University College London, London, United Kingdom, 2 Graduate School of Human Development and Environment, Kobe University, Kobe, Japan, 3 Department of Statistical Science, University College London, London, United Kingdom, 4 Center for Integrative Ecology, School of Life & Environmental Sciences, Deakin University, Victoria, Australia

* gandon.enora@gmail.com

**Data Availability Statement:** All relevant data are within the paper and its Supporting Information files.

**Funding:** EG This project has received funding from the European Union's Horizon 2020 research

## Abstract

Studies have documented that traditional motor skills (i.e. motor habits) are part of the cultural way of life that characterises each society. Yet, it is still unclear to what extent motor skills are inherited through culture. Drawing on ethnology and motor behaviour, we addressed this issue through a detailed description of traditional pottery skills. Our goal was to quantify the influence of three kinds of constraints: the transcultural constraints of wheel-throwing, the cultural constraints induced via cultural transmission, and the potters' individual constraints. Five expert Nepalese potters were invited to produce three familiar pottery types, each in five specimens. A total of 31 different fashioning hand positions were identified. Most of them (14) were cross-cultural, ten positions were cultural, five positions were individual, and two positions were unique. Statistical tests indicated that the subset of positions used by the participants in this study were distinct from those of other cultural groups. Behaviours described in terms of fashioning duration, number of gestures, and hand position repertoires size highlighted both individual and cross-cultural traits. We also analysed the time series of the successive hand positions used throughout the fashioning of each vessel. Results showed, for each pottery type, strong reproducible sequences at the individual level and a clearly higher level of variability between potters. Overall, our findings confirm the existence of a cultural transmission in craft skills but also demonstrated that the skill is not fully determined by a cultural marking. We conclude that the influence of culture on craft skills should not be overstated, even if its role is significant given the fact that it reflects the socially transmitted part of the skill. Such research offers insights into archaeological problems in providing a representative view of how cultural constraints influence the motor skills implied in artefact manufacturing.

## Introduction

Cultural diversity is observable in most human complex behaviours and their outcomes, including craft skills and artefacts [1–3]. Studies in ethnology and anthropology have long

and innovation programme under the Marie Sklodowska-Curie grant agreement No 793451 The funders had no role in study design, data collection and analysis, decision to publish, or preparation of the manuscript.

**Competing interests:** The authors have declared that no competing interests exist.

since documented that traditional motor skills (i.e. motor habits), such as the way people sit, swim, walk, carry loads, and so on, are part of the cultural way of life that characterises societies [4–8]. Yet, it is still unclear to what extent, for any given task, traditional motor skills are inherited through culture. We do not know how much cultural transmission—via the social guidance occurring through motor learning—determines the skills transmitted through generations. This lack of knowledge limits the understanding of cultural transmission, depriving archaeologists of a valuable framework for interpreting artefact variability.

Within archaeological research, it is accepted that motor skills used in the fashioning of pottery are more resistant to change than the skills used in other operations of the *chaîne opératoire* such as material preparation or artefact decoration [9–12]. Motor skills used in fashioning are highly specialized, developed through several months or years of extended deliberate practice supported by social guidance [13]. These fashioning skills would carry the specificity of the social context where they have been learnt, even when adult craftsmen move geographically [10]. The goal of the research we present here is to highlight and measure the cultural mark imprinted on specialized fashioning skills during apprenticeship. In doing so, we wish to provide archaeologists with a representative view of how cultural constraints influence the motor skills implied in artefact manufacturing.

Fig 1 presents the behavioural mechanisms of motor skills and of their cultural transmission over the scale of a community of practice [14]. Motor skills are described as behaviours

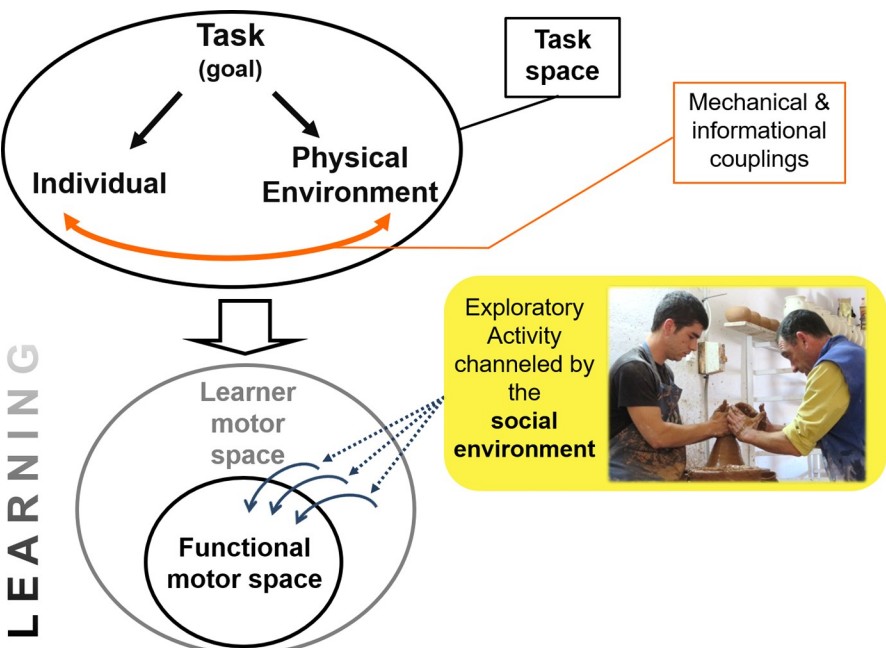

**Fig 1. Motor skills and their cultural transmission.** Motor skills (i.e. functional motor space and learner motor space) result from the interplay of three factors that together define the task space: the ergonomic and behavioural aspects of individual, the task, and the physical environment. The task has a structuring role on the individual behaviour (because it is oriented to reach a specific goal) and on the organisation of the physical environment (with the presence of typical tools and materials, finished vessels, etcetera). The individual interacts with his physical environment through real-time informational and mechanical couplings. The set of the different equivalent motor skills for a given task is called the functional motor space. Before reaching functionality through the learning process, the movements developed by the learner constitute the learner motor space. During the learning, the exploratory activity of the learner is channelled by the social environment. This social guidance constitutes the cultural transmission of motor skills and it supports the passage from the learner motor space to the functional motor space. Scheme adapted from [30, 31].

resulting from mechanical and informational couplings between the individual and his physical environment [15]. The adaptiveness of motor skills relies on a learning period during which the individual explores the *task space* to discover the sensorimotor information used to perform the task [16–18]. Importantly, developing a motor skill is not an ex nihilo process but takes place into a given socio-historical context [19, 20]. Social guidance plays a role in motor learning through direct interactions with elders mastering the task, or through the structuration of the learner practice. Thereby, the learner's attention is channelled so that his/her exploratory activity occurs over an optimal area of the *task space* [21–25]. This social channel corresponds to the *cultural transmission of motor skills* [26], which can be considered as cultural constraints influencing motor skills in addition to the task space constraints and the individual constraints. The social guidance facilitates the learning and paves the way for the development of a culturally specific skill [1, 27, 28]. Given the neuromuscular redundancy of the human body [29], a task can be achieved by several equivalent motor skills constituting the *functional motor space*. It is this latitude in motor skill execution that generates a cultural variability patterning, each society's 'style' of skill representing a subset inside the *functional motor space*.

We set up a field experiment with expert craftsmen for assessing the cultural influence on craft skill on the basis of empirical data. As a culturally wide-spread traditional craft, pottery wheel-throwing was chosen as an appropriate model [3]. Starting with a formless lump of clay, the goal of wheel-throwing is to produce a vessel, of a shape chosen in advance, using a wheel rotating in the horizontal plane at speeds varying between 50 and 150 rotations per min [32–34]. The difficulty of the task reflects in the vessel geometry (size and shape) and in the mechanical constraints inside the vessel (determined by the clay properties, the wall thicknesses, and the pot geometry) [35, 36]. In fashioning the vessel, potters successively deploy several distinctive hand positions for contact with the clay (Fig 2) and a given hand position can be used at different moments during the fashioning process [37–39]. The potter's hand positions constitute the observable aspects of wheel-throwing skill. We have noticed from our ethnographic observations among different worldwide pottery-making communities that tutors encourage their novices to watch their positions carefully so that they can reproduce them themselves. These hand positions would thus constitute an essential element of the skill transmitted through the apprenticeship.

The experiment involved professional Nepalese potters who were asked to throw a series of pottery types in their familiar conditions of practice. The successive hand positions used by the

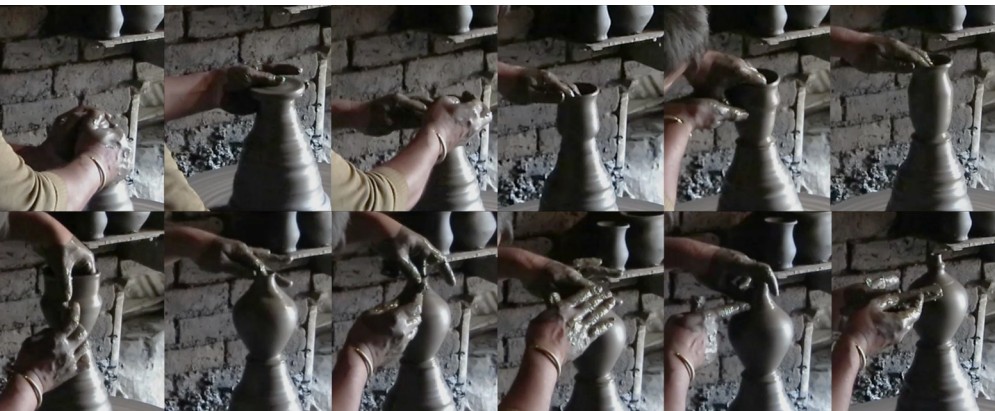

**Fig 2. Examples of hand positions used during wheel-throwing.** The images were extracted from a video recording of a Nepalese potter throwing a Money-Bank.

potters in the fashioning process were video recorded in controlled conditions in order to be systematically coded with timing software. Our goal was to quantify the influence of the three kinds of constraints on the potters' skills: the transcultural constraints of the wheel-throwing task, the cultural constraints induced via the cultural transmission, and the individual constraints inherent to each potter. Due to the high level of constraints of wheel-throwing [35, 40] and due to the traditional aspect of such craft activity, we hypothesized the task constraints and the cultural constraints both equally affect the potters' skill and preponderantly to the individual constraints.

The core of our analysis focused on the hand position repertoires and sequences. Hand position repertoires are the sets of the different hand positions used by potters through the experiment. Hand position sequences correspond to the time series of the successive hand positions used throughout the fashioning of one vessel. We analysed the repertoires with reference to an already existing ethogram of 84 hand positions. This global ethogram was constituted with two Indian pottery-making communities of practice (Prajapati and Multani potters from Uttar Pradesh) and one French (experts from different regions of the country) [38]. We expected to identify new positions in the Nepalese pottery-making community of practice, together with cross-cultural positions indicating the influence of the transcultural task constraints. If new positions were identified, we sought to determine if they were shared between several participants (i.e. resulting from cultural transmission) or used only by one individual.

Hand position sequences were analysed in terms of within-potter and between-potter similarity. Within-potter similarity measures the behavioural reproducibility of each potter while between-potter similarity measures the homogeneity (similarity) of behaviours inside the group. We compared these two measurements to test our hypothesis of a prevalent cultural influence on potters' fashioning skill. A higher within-potter similarity would indicate that potters mainly behave in an idiosyncratic manner and have consistent individual styles. In contrast, a higher between-potter similarity would indicate that potters within a group behave similarly with minimal or no idiosyncrasies.

In line with Bleed [41], we argue that the motor behaviour approach can be fruitful for archaeologists in providing analytical tools, and theoretical knowledge built on previous experimental works. Our results will be discussed with some implications for archaeological attempts to use ceramic patterning to uncover social identity of manufacturers.

## Materials and methods

### Participants and cultural setting

The field experiment involved Newar potters (from the Prajapati Hindu caste) who were all living and working in the pottery neighbourhood of Bhaktapur (Nepal, Kathmandu district). Similarly to India, the pottery handicraft in Nepal is learned through vertical transmission within endogamous castes, which produce standardized traditional artefacts in mass production [40, 42, 43]. This social organization of production and apprenticeship is characterized by a high cohesion and conformism, which favors the homogenization of practices and products [44]. In the potters' community of Bhaktapur most of the family workshops are located around a public courtyard which is used as a drying area by most craftsmen. The Nepalese potters traditionally used a stick-wheel (like the Prajapati Indian potters), but now they work with an electrical wheel (adopted since 2009). Five expert potters (all right-handed men) gave their written consent to participate in the study: LAX, SAN, RAM, SHI, and DIN. They were all over 25 years old (Mean ± SD: 38.0 ± 8.7 yrs) and had a minimum of ten years of wheel-throwing experience (26.0 ± 8.9 yrs). They previously either worked in the same pottery workshop

(LAX and SAN) or in neighboring workshops in the same street. SAN and DIN learnt the skill from their father LAX, RAM and SHI are SAN and DIN's uncles.

## Experimental protocol

For the purposes of the experiment, the participants were invited to produce three familiar medium sized pottery types: *Anchora* (vessel containing water used in rituals), *Money-Bank* (piggy bank) and *Ashtray* (ashtray) (Fig 3). As in their normal conditions of practice, potters relied on their practical experience of the pottery type during the experiment and no visual model was presented. Hence, each participant was free to produce the pottery types following his own usual fashioning way. To assess the reproducibility of behaviour, each pottery type was produced in five trials, thus each potter producing a total of 15 vessels. All participants performed this experimental task in SAN's workshop, using the same wheel and the same sort of clay. Participants used a basin of water to wet the clay, and a wooden scraper during the thinning and/or final fashioning steps. Potters produced the 15 vessels in a single session working at their own pace.

## Data recording and analysis

The fashioning of all vessels was video recorded (Canon PowerShot-SX270) and the finished products were photographed after drying to be analysed [45]. The video recordings of the potters' gestural patterns were systematically coded with the timing software Boris® [46]. An identifier (a number) was attributed to each hand position used by the potters through the fashioning. We captured not only the order of the successive hand positions, but also their temporal location and duration (in seconds). Data extracted from these videos were analysed in three stages, which are detailed below.

## Fashioning duration and number of fashioning gestures

In the first stage of the analysis we focused on the fashioning duration (in seconds) and the number of fashioning gestures, two global measures of the participants' behaviour. A fashioning gesture corresponds to the action of exerting pressures onto the clay by placing the hands on it (given certain positions) and by moving the hands through space from the base of the pot to its top. In our study, the number of fashioning gestures corresponds simply to the number of hand positions in each trial. The fashioning duration was measured, for each trial, from the beginning of the first gesture to the end of the last gesture.

## Hand position repertoires

In a second stage of analysis, we concentrated on the hand position repertoires. Part of the hand positions we identified already existed in the global ethogram, which was based on

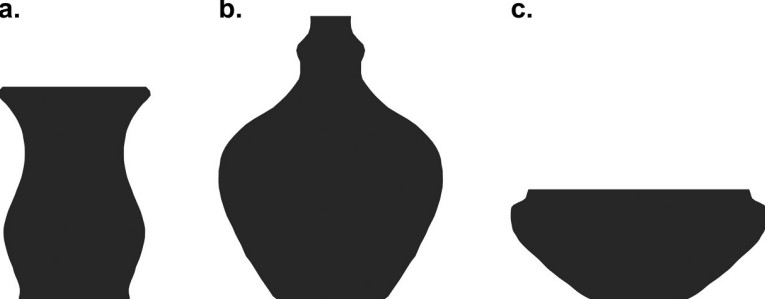

**a.**        **b.**        **c.**

**Fig 3. Traditional pottery types produced in the experiment.** Anchora (a), Money-Bank (b), and Ashtray (c).

French, Indian Prajapati, and Indian Multani potters [38]. We labeled those positions 'cross-cultural' and coded them using their corresponding identifier from the existing ethogram. The other positions identified were new and we added them in the global ethogram with a given identifier, a descriptive text and several photos. We labeled these new positions either 'cultural' (used by at least two participants), 'individual' (used by only one participant), or 'unique' (used only one time through the whole experiment and by one participant). After having established this characterization, we analysed the frequency of each hand position dependent on the different participants and pottery types. Finally, we evaluated the repertoire size which corresponds to the total number of different hand positions used by a potter in the whole experiment.

Having established their respective normality, we ran ANOVAs (with post-hoc paired t-tests) on the fashioning duration, number of fashioning gestures, and repertoire size. This evaluates the effect of pottery type (Anchora, Money-Bank, Ashtray). ANOVAs results are presented in the result section and the post-hoc results are presented in S1 Table. We also tested for differences in hand positions used by the four cultural pottery-making communities (French, Indian Prajapati, Indian Multani, and Nepalese) using a Fisher-Freeman-Halton exact test (Monte Carlo exact test) on the numbers of each hand position used in each culture (S2 Table).

## Hand position sequences

In the last stage of data processing, we analysed the hand position sequences defined as the time series of the successive hand positions used throughout the fashioning. All hand position sequences were re-scaled in time (maximum time set to 100) so that the sequences of different trials could be aligned and compared. These re-scaled hand position time series were subjected to cross recurrence quantification analysis (CRQA), a method classically used to quantify shared activity between two time series. CRQA starts with the formation of a cross recurrence plot [47, 48]. The plot is essentially a matrix that shows the temporal relationship between all possible combinations of hand positions in one trial with hand positions in another trial during the fashioning. In this plot, diagonal lines correspond to the sequences of hand positions that are shared between the pair of trials (Fig 4). The length of the longest diagonal line in the cross recurrence plot (called $L_{max}$) reflects the length of the longest shared sequence of hand positions between the two trials, and provides a measure of the similarity of the sequential pattern of hand positions. Note that this index of similarity $L_{max}$ reflects not only the similarity of hand positions used, but also the similarity of their temporal order. The higher the value of $L_{max}$, the more similar the pair of sequences. We computed $L_{max}$ for every combination of trial pairs using a customized Matlab routine [49]. The mathematical methods of computing $L_{max}$ in CRQA have been described in detail elsewhere [50]. To determine whether hand position sequences from the same potter were more similar to each other than to those from different potters, we then compared the $L_{max}$ values from within-potter pairs and between-potter pairs using unpaired t-tests with a method for unequal sample size and unequal variances [51]. For support, we present Cohen's ds as effect sizes.

## Ethics statement

The study consisted of non-invasive behavioural observations of potters in their familiar conditions of practice. Potters gave informed written consent prior to participation and were paid for their participation according to the local rates of the profession. These observations were made in the framework of a post-doctoral project of Aix-Marseille University (France). According to the current French laws on the protection of persons in biomedical research (law

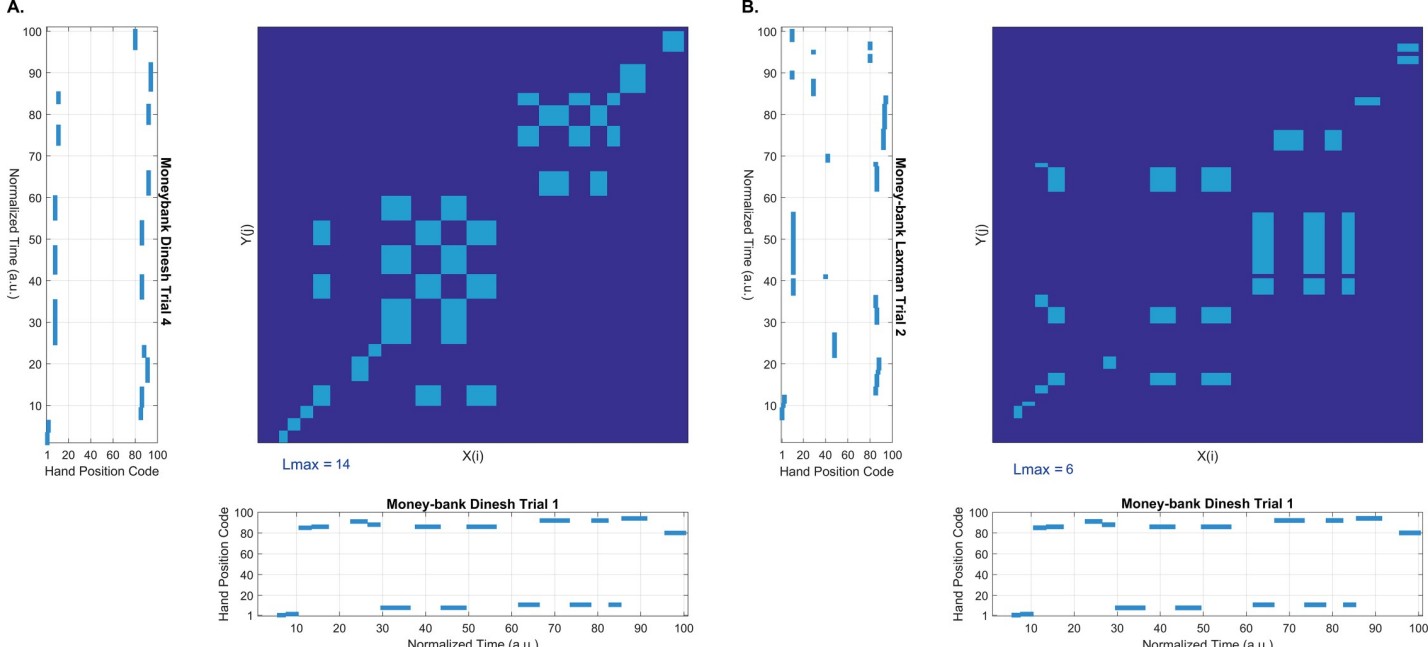

**Fig 4. Example of Money-Bank cross recurrence plots.** A: pair of sequences from one potter (DIN); B: pair of sequences from two different potters (DIN and LAX). Light blue regions indicate when the same hand positions occurred in the two sequences which are shown alongside the horizontal and vertical axes. The index of similarity, $L_{max}$, is indicated below left on each plot. It corresponds to the longest shared sequence pattern between the pair of hand position sequences.

No 88–1138, so-called Huriet-Serusclat law of the 20th December 1988, amended in 2004—law of the 9th August 2004), such a protocol does not require the approval of an ethics committee. The study complies with the ethics guidelines provided by the National Consultative Ethics Committee of the French Centre National de la Recherche Scientifique (COMETS). Participants were given 3-letters names. The cultural identification of the participants was simply their respective home countries.

## Results

### Fashioning duration and number of fashioning gestures

The average fashioning duration was less than one minute (47.8 s) for all three pottery types and five potters combined (Fig 5, top panel). Fashioning duration was significantly different among the three pottery types, the Money-Bank was the longest to produce (64.8 s), then the Ashtray (44.0 s), and finally the Anchora (34.6 s) ($F_{(4, 8)} = 13.68$; $p = 0.003$). The average number of fashioning gestures to produce a vessel was 16.4. This number was also influenced by the pottery types, potters deployed more gestures for the Money-Bank (19.4), then for the Ashtray (17.0), and for the Anchora (12.7) ($F_{(4, 8)} = 13.22$; $p = 0.003$) (Fig 5, bottom panel). Visual observation of Fig 5 shows similar individual differences on the fashioning duration and on the number of gestures. Considering the means across the three pottery types, we observed that SAN was faster (35.3 s) and used the fewest gestures (12.8), RAM and DIN took slightly longer (43.4 s and 43.9 s respectively) and used more gestures (13.7 and 15.4 respectively), and finally LAX and SHI took the longest (59.7 s and 56.5 s respectively) and used the most gestures (20.9 and 19.1 respectively). The small error bars displayed on the two graphs indicate the high reproducibility of each potter's performance for the two global behavioural variables.

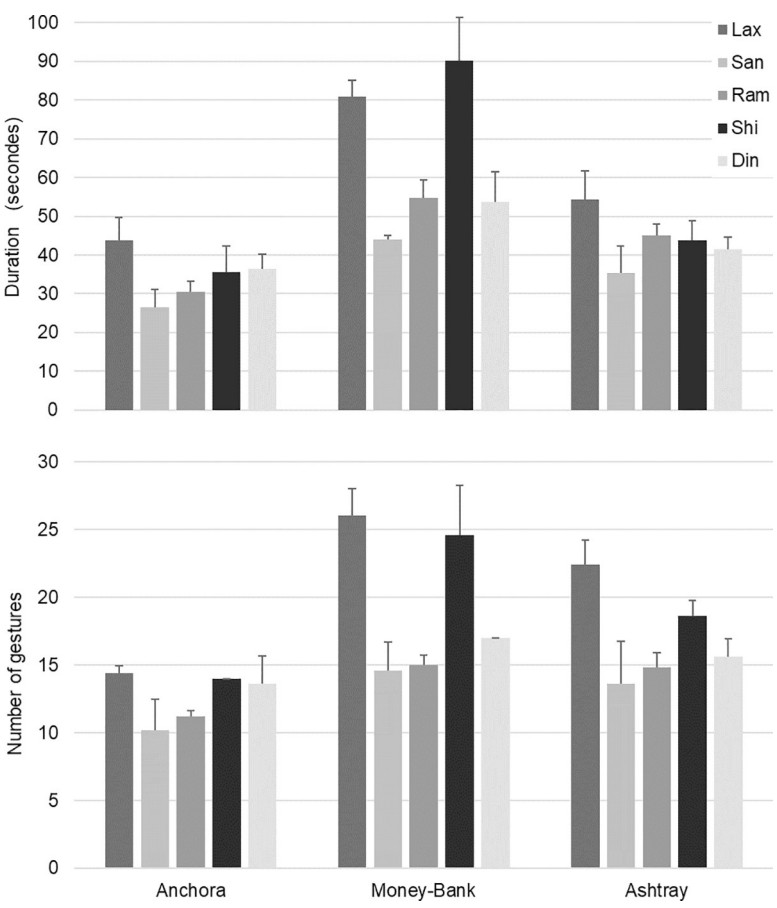

**Fig 5. Mean duration of the fashioning and mean number of fashioning gestures.** Top panel: duration (in seconds); bottom panel: number of gestures. The three pottery types are presented separately. Error bars indicate the standard error from the five trials of each participant.

## Hand position repertoires

A total of 31 different hand positions have been identified across the three pottery types and five participants. Most of them (14) were cross-cultural, ten positions were cultural, five positions were individual (DIN is the only potter who did not use an individual position), and two positions were unique (Figs 6 and 7). Hence, the global ethogram across the four cultural pottery-making communities of practice (French, Indian Prajapati, Indian Multani, and Nepalese) reaches 101 hand positions, 84 positions through the French and the two Indian groups in addition to the 17 Nepalese positions (10 cultural, 5 individual, and 2 unique). The subsets of hand positions used by each of the four communities are presented in S2 Table. Clear differences appeared among these subsets and the probability of getting these differences by chance was close to zero (Table 1). This provides ample evidence that the Nepalese subset of hand positions is distinct from those of the three other communities.

Fig 7 presents the frequency of the 31 hand positions used through the experiment. It summarises how each position has been used by each of the five potters for the three pottery types combined. The frequency varied significantly from one hand position to another, some positions (85 and 86) were used extensively (more than 25 times), while others (12, 89, 101) were used only once. Twelve positions were used by all potters: positions 85 and 86 (used on average 25 times or more), positions 11 (used over 20 times), positions 8, 80, and 88 (used 15 times or

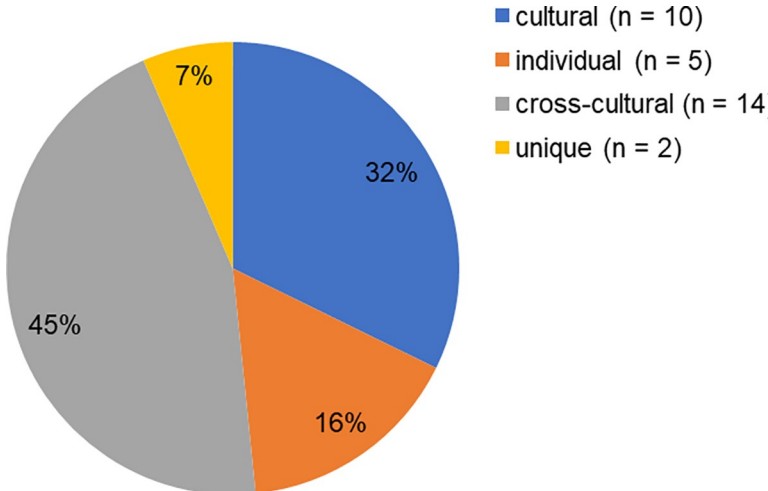

**Fig 6. Cross-cultural, cultural, individual, and unique hand positions.** Percentages of the different types of hand positions used through the experiment, for all three pottery types and five participants combined (absolute values in parentheses).

more), positions 1, 2, and 97 (used over 10 times), and positions 70, 92, and 99 (used over 5 times). A half of these 12 'shared' positions are cultural (positions 85, 86, 88, 92, 97, and 99), the other half are cross-cultural (1, 2, 8, 11, 70, and 80). From the total repertoire of 31 hand positions, 12 positions are 'basic' in the sense that they are used interchangeably for the three pottery types. In contrast, 12 hand positions are 'specific', as they are used solely for one pottery type: six positions (42, 92, 93, 94, 95, and 98) are specific to the Money-Bank, five positions (90, 96, 97, 99, and 100) are specific to the Ashtray and only one position (70) is specific to the Anchora (S1 Fig).

For all three pottery types combined, the mean repertoire size for each potters' hand positions was 19 (± 3.7), with a minimum of 15 (DIN) and a maximum of 25 (LAX) (S3 Table).

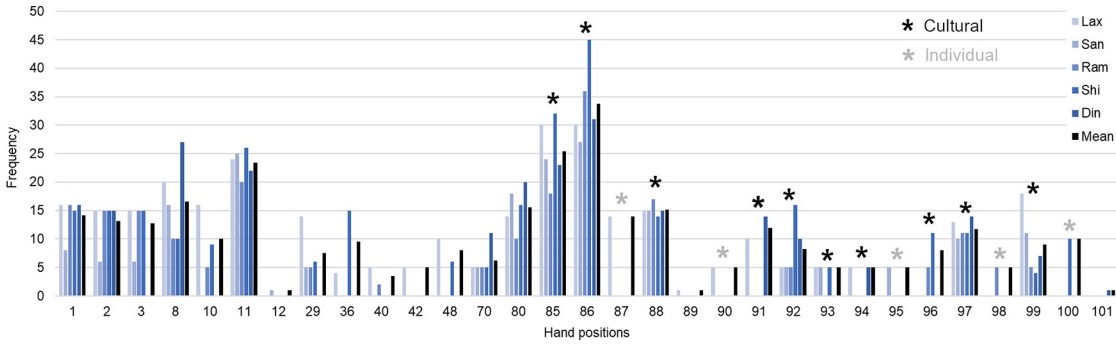

**Fig 7. Frequency of each different hand position used through the experiment.** Data for the five participants separately and the three pottery types combined. On the x-axis each position is noted by its identifier (ethogram number). For each participant, the frequency of a given hand position corresponds to the total number of times this position was used through the 15 fashioning sessions (3 pottery types x 5 trials). The mean (in black) corresponds to the average of the frequencies across the five participants. The first 14 hand positions from the origin of the x-axis (1–80) are cross-cultural, all the following 17 hand positions (85–101) are either (i) cultural (used by all 5 participants: 85, 86, 88, 92, 97, 99; or used by at least 2 participants: 91, 93, 94, 96), (ii) individual (87, 90, 95, 98, and 100) or (iii) unique (89, 101). The transcultural hand position 12 has been used only once by SAN, hence it is an unique position in the Nepalese group involved in our study but a common position for French, Indian Prajapati, and Indian Multani potters (S2 Table). The hand position 42 has been used only by one participant here (LAX), but it has also been used by two Indian Multani potters (S2 Table).

Table 1. Results of the Fisher-Freeman-Halton exact test on the hand positions by cultures. This tests for differences in hand positions used by the four cultural communities (French, Indian Prajapati, Indian Multani, and Nepalese) (S2 Table).

|  | Fisher Statistic | Probability |
|---|---|---|
| All 4 cultures | 477.4 | 1.185e-181 |
| Nepalese vs. French | 150.2 | 1.736e-53 |
| Nepalese vs. Prajapati | 157.0 | 1.859e-54 |
| Nepalese vs. Multani | 138.9 | 1.794e-50 |

This repertoire size is close to the one reported in traditional Indian potters producing three customary medium sizes vessel types (21.3 ± 4.4) [39]. As for the fashioning duration and the number of gestures, the repertoire size was influenced by the pottery types, the Money-Bank requiring the largest repertoire (13.1), then the Ashtray (11.8) and Anchora (9.6) ($F(4, 8)$ = 38.81; $p < 0.001$) (S2 Fig).

## Hand position sequences

The hand position sequences were analysed in terms of similarity using the index $L_{max}$ [50]. Fig 8 shows the average $L_{max}$ values of within-potter and between-potter pairs for each pottery type separately. We observed that the $L_{max}$ values of within-potter pairs were significantly higher than those of between-potter pairs for each pottery type (Fig 8 and Table 2). This demonstrates that when fashioning the pots, participants vary more between each other than within themselves (see S3 Fig for representative hand positions sequences plots). The high individual reproducibility reported here, and also previously in the global behavioural variables (Fig 5, S2 Fig), corroborates the significant individual standardization of the final vessels thrown by the five participants [45].

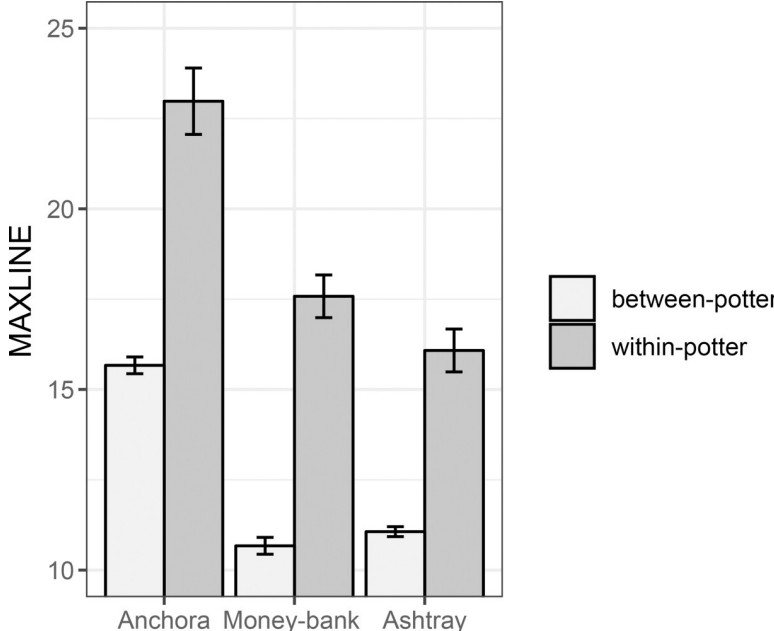

Fig 8. $L_{max}$ values computed within-potter and between-potter. The higher the index, the more similar the compared sequences. The three pottery types are presented separately. Error bars represent the standard errors of the means.

**Table 2. Unpaired t-test results and associated Cohen's d values, for each pottery type (Anchora, Money-Bank, Ashtray).** This test compared $L_{max}$ values between hand positions sequences from within-potter pairs and between-potter pairs.

|  | t | df | P | Cohen's d |
|---|---|---|---|---|
| Anchora | 7.72 | 55.48 | <0.0001 | 1.71 |
| Money-bank | 10.87 | 65.21 | <0.0001 | 1.83 |
| Ashtray | 8.22 | 54.34 | <0.0001 | 1.92 |

## Discussion

The use of an experimental approach grounded in motor behaviour provides new insights for exploring the influence of culture on craft skills. First, our findings confirmed that acquiring a craft skill within a community of practice indeed leads to a cultural marking [1, 27, 28]. A Nepalese motor tradition appeared through 10 specific cultural hand positions (Fig 6) and through a culturally distinctive subset of hand positions (S2 Table, Table 1). At the same time, our results demonstrated that the craft skill is not fully determined by a cultural marking. When considering each behavioural variable analysed in our study, one is struck by the impact of either the transcultural task constraints or the individual constraints.

The transcultural task constraints were revealed through the prevalence (43%) of cross-cultural hand positions inside the total repertoire collected through the video recordings (Fig 6). Hence, most of the positions we identified in our experiment with Nepalese potters are also used by Indian Prajapati, and/or Indian Multani, and/or French potters. Interestingly, we noticed that cross-cultural positions were overrepresented in the first fashioning phase (centering and hollowing) in which the shape parameter does not come into play. It seems logical that potters originated from distinct cultural settings used common hand positions due to the universally shared constraints of the physical ergonomics inducing limited options in the way to use human hands. It is very probable that a collection of all worldwide potters' hand positions would contain solely cross-cultural positions at least for the initial stages of fashioning. Therefore, the cultural influence on potters' repertoires cannot be explicitly signalled by solely one or two positions but by a distinctive subset of positions used amongst several craftsmen of a community. Further studies will be required to evaluate how these cultural subsets of positions are distinguished at different social scales (family, community, village, region).

We also interpret the restricted range of values measured for each global variable as an influence of the transcultural task constraints. Throwing medium-size clay pots with a high-speed wheel, all participants fall into equivalent performances in terms of fashioning duration, number of fashioning gestures, and hand positions repertoire size (Fig 5, S2 Fig, S3 Table). The performances presented in our study were comparable to the ones collected with Indian craftsmen producing similar medium-sized artefacts [39]. One is inclined to think that the weight of the task constraints determines the latitude in motor skills execution, and the associated cross-cultural traits. For highly constrained tasks such as wheel-throwing the different subsets of cultural skills (developed in distinct communities) largely overlapped inside the *functional motor space*. Yet, whatever the pervasiveness of the task constraints, it seems not to exclude completely latitude in motor skills execution, and possibilities for variations. As observed in our results, even though wheel-throwing fashioning is highly constrained, there is still room for cultural and individual variations.

A strong result of this work is the significant effect of the individual constraints on the potters' motor skill. This was observed at first in the 16% of individual hand positions inside the total repertoire (Fig 6). Although options in using hands are limited, surprisingly almost each

potter developed original positions. These individual positions do not result from the tutor's imitation but certainly from the individual exploration of the task space during the learning process. They could also result from the specificities of the individual physical constraints (i.e. physical ergonomics), such as the arm, hands, and finger dimensions, which could facilitate or prevent the novice to reproduce his tutor's hand positions. We also observed that participants varied in their personal repertoires, each of them having his proper 'toolbox' that comprised of a unique set of positions (Fig 7, S1 Fig). Moreover, not only did the positions differ amongst craftsmen, but also their order and temporal pattern in the fashioning sequence. We saw reproducible hand positions sequences at the individual level, and a clearly higher level of variability between potters, showing that each potter has his own preferred fashioning sequence that is distinct from those of others. Such behavioural idiosyncrasies were also exhibited in the fashioning duration, number of gestures, and hand positions repertoire size (Fig 5, S2 Fig, S3 Table). Taken together, these findings demonstrate that potters do not strictly reproduce a given cultural fashioning pattern. This was most clearly exhibited through the differences between the skill of LAX and those of his two sons SAN and DIN (Figs 5 and 7, S3 Fig, S3 Table). Following Bril [30], we suggest that the social learning of craft skills is not truly to imitate the elders' skill but to be guided by social pointers towards the individual discovery of any movement effective for the task. The fact that craftsmanship requires skills and expertise well established in the knowledge of artisanal rules does not mean that there is no room for diversity of individual virtuosity. The developed skills are therefore likely to contain both cultural and individual traits. While interpreting archaeological records, it is a fact important to bear in mind that cultural and individual influences cohexist within human motor behaviour and artefacts [52]. The next step in cultural transmission research will be to capture the hand positions in several expert and novice couples, so that we can measure systematically the variation occurring through learning.

Before discussing more precisely the archaeological implications of our work, it seems necessary to tackle an underlying issue raised by our results: the links between the fashioning behaviours and their material results (i.e. vessels' profiles). The performances of the participants revealed that Money-Bank was the more difficult to throw and Anchora the least (Fig 5, S2 Fig). This can be explained by the differences in geometries and mechanical constraints. The Money-Bank is bigger than the two other types, but above all, it is a closed shape with higher risk of collapse. The slightly higher difficulty for the Ashtray is due to its rim that requires some finishing operations, in contrast to the Anchora. Thus, potters do adapt their fashioning behaviours to the requirements of the final shape. This adaptation was also visible in the specific hand positions used for the Money-Bank, Ashtray, and Anchora (six, five, and one specific hand positions respectively) (S1 Fig). The use of pottery-type specific positions was similarly shown in Indian potters fashioning behaviours [39]. One might infer from those results that the hand positions determine the final vessel profile, with certain pottery types being easier to fashion using certain hand positions. But this reasoning is in fact fallacious because we have presented empirical evidence of the functional equivalence of totally distinct hand positions [37]. The plastic deformations of the clay result from the fine tuning of the manual pressures which are not determined by the hand positions. This is not, of course, to say that the hand positions do not play a role in the final shape, as potters have certainly learnt to produce the different types of vessels using some preferentially positions. As illustrated in our study, in a given community of practice certain hand positions are associated with particular pottery types. In our view, the nature of this association is contingent on socially transmitted potters' preferences. Following this interpretation, we argue that the individual signatures reported in the geometry of the final vessels produced by the five participants [45] are not the direct result of the individual hand positions reported in the present contribution, but the

result of the manual pressures which merit further exploration. The only thing we can say here is that the individual signatures on pots' profiles are associated with individual fashioning behaviours in terms of fashioning duration, number of gestures, hand positions repertoire size, hand positions and their sequences.

By focusing on the observable behaviours of traditional potters our results reveal cultural and individual traits corresponding to cultural and individual technical styles [10, 53–56]. Since they are embedded in automated skills developed through long lasting socially guided apprenticeship, these styles are resistant to change and intimately related to the social identity of the practitioners: *"[. . .] technical style offers an opportunity to explore the deepest and more enduring facets of social identity."* (Gosselain, 1998:82). One may ask if we can infer from ancient pots the technical style and identity of their producers, and by extension the spatial and temporal distribution of societies. As we explained above, the potters' hand positions did not imprint univocal traces in the ensuing vessels. Hence, at least for the elementary actions of a given fashioning technique, the technical styles constitute nonmaterial aspects of culture which are difficult to approach through archaeological research. The traces on ancient pots can simply distinguish the different fashioning techniques used to produce a vessel (wheel-throwing and wheel-coiling for example) [57, 58]. Interpreted in relation to other body of evidence (dating, sites location, petrography), these fashioning techniques can contribute to identify social boundaries [59] and cultural lineages [60].

The main contribution of our work to guide the exploration of ancient artefacts lies in the recognition of the roles played by the different constraints on craft skill. The task constraints provide information about the latitude of possible cultural and individual variations in craftsmen behaviours. Then, if there is evidence that vessels come from a given community of practice, one should be aware that vessels falling into homogeneous geometrical types are not produced by culturally standardized fashioning patterns but by behaviours containing both cultural and individual features. The pool of individual behavioural variants existing in a community of practice would constitute a possible source of change in artefacts.

## Conclusion

In supporting the apprenticeship of craft skills, cultural transmission orients the apprentice toward the learning of a cultural skill, a skill with certain characteristics that are specific to the apprentice's community of practice [1, 27]. This cultural skill can be considered as a 'model' shared by the community members and transmitted across generations. So far, the true impact of this cultural model on craftsmen behaviour remains largely unexplored. Drawing on ethnology and motor behaviour, we addressed this issue through a detailed description of traditional potters' fashioning skill. As expected, we discovered a significant cultural influence through culturally specific hand positions, indicating that participants kept a mark of their learning guided by elders. However, our overall results demonstrated that the skills developed by the potters not only reflect a cultural marking, but also include both the transcultural task constraints and the individuals' singularities. Hence, the influence of culture on craft skills should not be overstated, even if its role is significant given that it reflects the socially transmitted part of the skill closely linked to the identity of practitioners.

We can assume that the individual and cultural influences on the manufacturing behaviour of craftsmen depend on the cultural context, notably because models of learning vary culturally [20, 61, 62]. The more that learners are given responsibility and autonomy, the more diverse would be the fashioning patterns in a community of practice (with a significant number of individual hand positions). In contrast, if tutors explicitly insist that their novices accurately reproduce their gestures, then a given set of cultural hand positions can perpetuate in

lineages of craftsmen. To test these hypotheses requires future research analysing concurrently the potters fashioning behaviours and the models of learning (through the teaching interactions between tutors and their novices).

Such research plays a crucial role in the generation of empirical data on craftmen motor skill and furnishes results which offer insights into archaeological problems. In particular, one would encourage archaeologists, in their attempt to interpret ancient artefacts, to recognize the different constraints acting on motor skill, and to weight their relative impact based on all contextual information available, notably the level of specialization of the task and the context within which learning, production, and distribution has taken place.

## Supporting information

**S1 Fig. Frequency of each different hand position used throughout the experiment separated by pottery type.** Bottom panel: Ashtray, middle panel: Money-Bank, top panel: Anchora. The graphs present the data of the five participants separately. On the x-axis each position is noted by its ethogram number. For each participant and each pottery type, the frequency of a given hand position corresponds to the total number of times this position was used throughout the five fashioning sessions (i.e. five trials).
(PDF)

**S2 Fig. Size of the hand position repertoires.** This graph shows the total number of different hand positions used by each potter for each pottery type separately. Error bars indicate the standard error from the five trials of each participant.
(TIF)

**S3 Fig. Representative plots of hand position sequences.** The two first plots present the sequences of SAN and his father LAX producing an Anchora (first trial). The five following plots present the sequences of DIN (also son of LAX) producing the five Anchora (five trials).
(PDF)

**S1 Table. Paired t-test post-hoc results for the fashioning duration, number of gestures, and repertoire size.** We ran ANOVAs on the fashioning duration, number of fashioning gestures, and repertoire size (see the results in the manuscript). To identify significant differences between the three pottery types, we ran paired t-tests post-hoc with Tukey multiple testing correction.
(DOCX)

**S2 Table. Hand positions by cultures.** The first column contains the hand positions identifiers (numbers 1 to 101). The four following columns contains the number of potters from each group (French, Prajapati, Multani, and Nepalese) using the different positions.
(XLSX)

**S3 Table. Repertoire size of the hand positions for each potter, for all three pottery types combined.** This repertoire represents all the different positions used by a potter in the whole experiment. Between-potter mean and standard deviation are indicated in the two last columns.
(DOCX)

**S1 Data.**
(ZIP)

## Acknowledgments

We are grateful to the potters who participated in the experiments, particularly to Sanjay Prajapati for welcoming us in his workshop. We also wish to thank Olivier Friard for providing technical assistance with the data processing, Reinoud Bootsma for his support during the experiment, Sébastien Manem for his advice with the bibliography, and the two reviewers for helpful comments.

## Author Contributions

**Conceptualization:** Enora Gandon.

**Data curation:** Enora Gandon, Tetsushi Nonaka, John Endler.

**Formal analysis:** Enora Gandon, Tetsushi Nonaka, Raphael Sonabend, John Endler.

**Funding acquisition:** Enora Gandon.

**Investigation:** Enora Gandon.

**Methodology:** Enora Gandon, Tetsushi Nonaka, John Endler.

**Supervision:** Enora Gandon.

**Validation:** Tetsushi Nonaka.

**Visualization:** Enora Gandon.

**Writing – original draft:** Enora Gandon.

**Writing – review & editing:** Enora Gandon, Tetsushi Nonaka, Raphael Sonabend, John Endler.

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
