## [Decision Letter · Decision Letter 0]

30 Jun 2020

PONE-D-20-08524

Assessing the influence of culture on craft skills: Quantitative study with expert Nepalese potters

PLOS ONE

Dear Dr. Gandon,

Thank you for submitting your manuscript to PLOS ONE. After careful consideration, we feel that it has merit but does not fully meet PLOS ONE’s publication criteria as it currently stands. Therefore, we invite you to submit a revised version of the manuscript that addresses the points raised during the review process.

We look forward to receiving your revised manuscript.

Kind regards,

Piotr Sorokowski

Academic Editor

PLOS ONE

Journal Requirements:

Reviewers' comments:

Reviewer's Responses to Questions

**Comments to the Author**

1. Is the manuscript technically sound, and do the data support the conclusions?

Reviewer #1: Yes

Reviewer #2: Yes

2. Has the statistical analysis been performed appropriately and rigorously? 

Reviewer #1: I Don't Know

Reviewer #2: Yes

3. Have the authors made all data underlying the findings in their manuscript fully available?

Reviewer #1: Yes

Reviewer #2: Yes

4. Is the manuscript presented in an intelligible fashion and written in standard English?

Reviewer #1: Yes

Reviewer #2: Yes

5. Review Comments to the Author

Reviewer #1: Although I cannot comment on the statistical analysis applied in this work, I do find that the content of the manuscript to be well-written, well-conceived, and intelligible. The data presented and the arguments made support the conclusions. This contribution raises and addresses important questions, and pushes the field further to allow for more nuanced interpretations of crafting behaviors. The approach described here has great potential for future applications in interpretation of human practices, particularly those which result from novice-expert relationships (and beyond the exclusive sphere of pottery production practices).

Despite this, there are two relevant missed opportunities here:

1. Discussion of observed potting gestures between father and son (if son did indeed learn from father), or other instance where vertical transmission would imply the novice-expert relationship which is described for future research in the discussion section.

2. Consideration of the ergonomic underpinnings of gestural variation. Some data collected relating to arm, hand, and finger dimensions may reveal potential grounds for deviation from learned gestures within novice-expert relationships. Could the ergonomic represent a further example of transcultural task constraint?

The authors describe potential for research into the long term gestural choices as a novice learns from an expert, and it is unclear whether any of the potters included in this study previously had this kind of relationship between them. If this is the case, the authors could provide some insight into the relationship between the gestures between potters who once had this novice-expert relationship in order to identify further meaningful gestural practices in their crafting behaviors. Presumably this can be included in the final manuscript without further need for fieldwork, and is recommended.

The consideration of ergonomic underpinnings for gestural variation is not addressed throughout the manuscript, but it would require further fieldwork (if not collected in initial fieldwork), a brief description of this variable is worth including. Future work on this topic would do well to consider the way that an individual’s physical dimensions can facilitate or restrict the effectiveness of particular gestures which are learned during novice-expert relationships.

Reviewer #2: This article presents an interesting addition for the discussion of craft skill. It will be of great interest to researchers that are engaged with Ethnographic studies, ceramic analysis, motor skill and more. The strength of the paper derives from the aim to highlight and measure the cultural mark imprinted on specialized fashioning skills during apprenticeship. The article aims to quantify the influence of three kinds of constraints (the transcultural constraints of the wheel-throwing task, the cultural constraints induced via the cultural transmission, and the individual constraints inherent to each potter). The authors conducted an experiment with five expert Nepalese potters, where they were asked to produce three traditional pottery types. They identified 31 different fashioning hand positioning, where most of them were classified into transcultural constraints. The results showed, for each pottery type, strong reproducible sequences at the individual level and a clearly higher level of variability between potters. In general, the current paper suggests that although there is a connection between cultural transmission to craft skills, the latter are not fully determined by a cultural marking and should not be exaggerated. These are very interesting results!

Yet I find problems in the manuscript and I would suggest that major revisions are necessary for it to be published in PlosOne:

1. Terminology: From my perspective several terms should be better clarified: the concept of ‘motor skills’ and what it entails is unclear, and many will be distracted on the overuse of the concept and its context. It is also worthy to explain what the authors mean by ‘fashioning operations’.

2. Figures:

- Figure 1 is problematic. I don’t understand it. Why is task equated to goal? The arrows are leading to directions which are not clear. Not all the relations are explained especially their location in the flow diagram.

- The experiment is conducted by expert Nepalese potters, why the hands/gestures from the Indian experiment are presented in Figure 2? I suggest that the authors will provide a new figure incorporating Nepalese gestures.

- In the text supplementary figures are inserted. I believe that eventually they will be included in the supplementary material. I suggest that some of the figures will be listed in the main article – pulled back to the main text.

- The absolute N values are missing in Figure 6. This is important since in the discussion these numbers are mentioned: “…through 10 specific cultural hand positions (Fig 6)”

- Finally, my most important comment related to Figure 7. This figure is an important presentation of the manuscript results. Yet, it is crowded with information. I suggest finding an alternative way to demonstrate/visualize these results – For example, in Gandon et al 2013 the cumulative percentage graph was harnessed. I’m sure there are many other ways to illustrate the distribution of gestures.

3. Methodology - It is recommended to add additional information on the potters. In particular point to family relations, village proximity etc.

Some of the sections in the methodology are not clear. In particular, I recommend adding information on the Maxline procedure. In addition, it will be very helpful if this section will be divided to sub-topics (sub-headings).

4. Results:

As mentioned above, this study managed to provide very interesting results, as, “… even though wheel-throwing fashioning is highly constrained, there is still room for cultural and individual variations.” Or “similar the products are given the high degree of individual gestures. “ or, “The developed skills are therefore likely to contain both cultural and individual traits (discussion, P.28). Yet isn’t this dependent on the cultural background. The variability between available cultural norms. Will we find the same individual signature in a culture with very strict cultural background? I suggest to tune this down.

In general, the results should be more straightforward since in some sections the paper finds that craft skills lead to a cultural marking, yet it is also stated that the craft skills in not fully determined by a cultural marking. This point needs to be clarified and extended in this regard.

5. The Archaeological viewpoint - It is not fully clear how this study can aid archaeologists in interpreting archaeological ceramic types. How is it possible to detect motor skills (that lead to cultural constraints) on the archaeological record? The authors state that it will be possible to identify traces throughout the vessel’s profiles (in case of fashioning techniques: wheel-throwing and wheel-coiling) – How? An archaeological example should be integrated to better clarify this point.

I recommend that the article will undergo final touch-ups in English editing

6. PLOS authors have the option to publish the peer review history of their article (what does this mean?). If published, this will include your full peer review and any attached files.

Reviewer #1: **Yes: **C.D. Jeffra

Reviewer #2: No

---

## [Author Response · Author response to Decision Letter 0]

21 Jul 2020

We are grateful to the reviewers for their constructive comments. As detailed below, we have taken into account all the points raised. The manuscript has been modified to make clearer the message of our article. We believe these revisions will considerably improve the potential impact of our contribution.

Reviewer #1:

1. Discussion of observed potting gestures between father and son (if son did indeed learn from father), or other instance where vertical transmission would imply the novice-expert relationship which is described for future research in the discussion section.

We have now added in the Method section that the two brothers SAN and DIN learnt the skill from their father LAX. We also clarified in the Discussion that the skill of LAX was distinct to those of his two sons SAN and DIN. This demonstrates that the skill is not perfectly copied by the learner during the cultural transmission of the skill. 

2. Consideration of the ergonomic underpinnings of gestural variation. Some data collected relating to arm, hand, and finger dimensions may reveal potential grounds for deviation from learned gestures within novice-expert relationships. Could the ergonomic represent a further example of transcultural task constraint? 

The authors describe potential for research into the long term gestural choices as a novice learns from an expert, and it is unclear whether any of the potters included in this study previously had this kind of relationship between them. If this is the case, the authors could provide some insight into the relationship between the gestures between potters who once had this novice-expert relationship in order to identify further meaningful gestural practices in their crafting behaviors. Presumably this can be included in the final manuscript without further need for fieldwork, and is recommended.

The consideration of ergonomic underpinnings for gestural variation is not addressed throughout the manuscript, but it would require further fieldwork (if not collected in initial fieldwork), a brief description of this variable is worth including.

Future work on this topic would do well to consider the way that an individual’s physical dimensions can facilitate or restrict the effectiveness of particular gestures which are learned during novice-expert relationships.

In line with your suggestions, we have now described in the Discussion that physical ergonomics could influence the potters’ fashioning behaviours and their transmission during the learning process. As answered in the first comment, we have now detailed the tutor-novice relationship between LAX and his two sons and the fact that their hand positions (repertoire and sequences) were different. 

Reviewer #2:

1. Terminology: From my perspective several terms should be better clarified: the concept of ‘motor skills’ and what it entails is unclear, and many will be distracted on the overuse of the concept and its context. It is also worthy to explain what the authors mean by ‘fashioning operations’.

We acknowledge that the concept of ‘motor skill’ is not widely shared in archaeology and may be confusing. This concept comes from the motor behaviour field where our study is grounded. We have now indicated in the Introduction that ‘motor skill’ is synonym of ‘motor habits’, a term which is more familiar in archaeology and anthropology. 

We have removed the word ‘operation’ to clarify the sentence (line 62). 

2. Figures:

- Figure 1 is problematic. I don’t understand it. Why is task equated to goal? The arrows are leading to directions which are not clear. Not all the relations are explained especially their location in the flow diagram.

We have detailed in the caption of Figure 1 why the task is considered as a ‘goal’ in the motor behaviour theoretical framework. The meanings of the different arrows have been made clear through additional sentences indicating the relations between the different parts of the diagram. 

- The experiment is conducted by expert Nepalese potters, why the hands/gestures from the Indian experiment are presented in Figure 2? I suggest that the authors will provide a new figure incorporating Nepalese gestures.

We now provide a Figure 2 with Nepalese hand positions.

- In the text supplementary figures are inserted. I believe that eventually they will be included in the supplementary material. I suggest that some of the figures will be listed in the main article – pulled back to the main text.

All the supplementary figures and tables which are indicated in the text are now listed at the end of the article.

- The absolute N values are missing in Figure 6. This is important since in the discussion these numbers are mentioned: “…through 10 specific cultural hand positions (Fig 6)”

We have added the absolute N values in the legend of the Figure 6 and indicated them in the caption of Figure 6.

- Finally, my most important comment related to Figure 7. This figure is an important presentation of the manuscript results. Yet, it is crowded with information. I suggest finding an alternative way to demonstrate/visualize these results – For example, in Gandon et al 2013 the cumulative percentage graph was harnessed. I’m sure there are many other ways to illustrate the distribution of gestures.

We are aware that it is not easy to extract information from Figure 7 which contains a lot of information. What we intend to show with this figure is the detailed frequency of each hand position, for each potter, underlying the positions which were cultural and those which were individual (by the black and grey asterisk). The cumulative percentage graphs could appear easier to grasp but in fact they are also misleading given their resemblance with time series data. They give to the reader the impression of hand position sequences and not the simple frequencies. Therefore, we decided to keep the figure as it is, adding details in the caption to help understanding. 

3. Methodology - It is recommended to add additional information on the potters. In particular point to family relations, village proximity etc. Some of the sections in the methodology are not clear. 

We now clearly state in the “Participants and cultural setting” sub-heading that the participants involved in the experiment were all living and working in the pottery neighbourhood of Bhaktapur (Nepal, Kathmandu district). We also indicate the family relations between the participants.

In particular, I recommend adding information on the Maxline procedure. In addition, it will be very helpful if this section will be divided to sub-topics (sub-headings).

The section about the hand position sequences analysis has been re-written with additional information on the Maxline index (renamed Lmax). We have also added sub-headings in the Method section to make it correspond with the Results section. 

4. Results: As mentioned above, this study managed to provide very interesting results, as, “… even though wheel-throwing fashioning is highly constrained, there is still room for cultural and individual variations.” Or “similar the products are given the high degree of individual gestures. “or, “The developed skills are therefore likely to contain both cultural and individual traits (discussion, P.28). Yet isn’t this dependent on the cultural background. The variability between available cultural norms. Will we find the same individual signature in a culture with very strict cultural background? I suggest to tune this down. In general, the results should be more straightforward since in some sections the paper finds that craft skills lead to a cultural marking, yet it is also stated that the craft skills in not fully determined by a cultural marking. This point needs to be clarified and extended in this regard.

We can assume, as suggested in this comment, that the individual and cultural influences on the manufacturing behaviour of craftsmen vary with the cultural context, notably because models of learning vary culturally (Rogoff, 1990, 1993). The more that learners are given responsibility and autonomy, the more diverse would be the fashioning patterns in a community of practice (with a significant part of individual hand positions). On the contrary, if tutors explicitly insist that their novices accurately reproduce their gestures, a given set of cultural hand positions can perpetuate in lineages of craftsmen. To test these hypotheses requires a concurrent analysis of the potters fashioning behaviours and the models of learning (through the teaching interactions between tutors and their novices). We have included this issue in the Conclusion to clarify the message of our article.

5. The Archaeological viewpoint - It is not fully clear how this study can aid archaeologists in interpreting archaeological ceramic types. How is it possible to detect motor skills (that lead to cultural constraints) on the archaeological record? The authors state that it will be possible to identify traces throughout the vessel’s profiles (in case of fashioning techniques: wheel-throwing and wheel-coiling) – How? An archaeological example should be integrated to better clarify this point.

We have revised the paragraph relating to the technological style to clarify the contribution of our study in the interpretation of archaeological vessels. We now refer to specific archaeological case studies. 

I recommend that the article will undergo final touch-ups in English editing

As recommended, the article has been revised for English language.

---

## [Decision Letter · Decision Letter 1]

1 Sep 2020

Assessing the influence of culture on craft skills: A quantitative study with expert Nepalese potters

PONE-D-20-08524R1

Dear Dr. Gandon,

We’re pleased to inform you that your manuscript has been judged scientifically suitable for publication and will be formally accepted for publication once it meets all outstanding technical requirements.

Kind regards,

Piotr Sorokowski

Academic Editor

PLOS ONE

Additional Editor Comments (optional):

Reviewers' comments:

Reviewer's Responses to Questions

**Comments to the Author**

1. If the authors have adequately addressed your comments raised in a previous round of review and you feel that this manuscript is now acceptable for publication, you may indicate that here to bypass the “Comments to the Author” section, enter your conflict of interest statement in the “Confidential to Editor” section, and submit your "Accept" recommendation.

Reviewer #1: All comments have been addressed

Reviewer #3: All comments have been addressed

2. Is the manuscript technically sound, and do the data support the conclusions?

Reviewer #1: (No Response)

Reviewer #3: Yes

3. Has the statistical analysis been performed appropriately and rigorously? 

Reviewer #1: (No Response)

Reviewer #3: Yes

4. Have the authors made all data underlying the findings in their manuscript fully available?

Reviewer #1: (No Response)

Reviewer #3: Yes

5. Is the manuscript presented in an intelligible fashion and written in standard English?

Reviewer #1: (No Response)

Reviewer #3: Yes

6. Review Comments to the Author

Reviewer #1: (No Response)

Reviewer #3: The study presented in the paper is interesting for several reasons. One of them is that the results it provides become invigorating for many other research fields. Another is that it presents an extremely innovative topic, so far poorly researched in archaeology.

After the authors received two anonymous reviews, they meticulously referred to most of the suggested corrections, terminological/technical and methodological alike (some of them were both detailed and collateral). It has been also revised for English language improvements.

Since the paper is written with methodological rigour and its contents shed light on the previously unexplored issue of motor skills (motor habits) in relation to traditional pottery, I recommend to publish it without any further corrections.

I do not share the concern of one of the reviewers, who suggests that the results should be more straightforward, since some of the claims made in Conclusion seem contradictory. The novelty of the subject matter undertaken by authors allows, in my opinion, such seemingly contradictory conclusions.

I believe that the presented study may be interesting not only for archaeologists, but also for researchers from, among the others, anthropological studies of art, cross-cultural and evolutionary aesthetics.

Let me explain it with a simple example. An attempt to investigate why specific crafts differs from each other gives in fact the opportunity to show what makes one craftsmanship more aesthetically pleasing (more beautiful, having a higher aesthetic value) than another. In this way, thanks to the study presented in the paper, the aesthetic concepts of skill and virtuosity (Dutton 2002) are given an interesting empirical explanation.

Aristotle has already divided art into techne and mimesis: the fact that techne requires skills and expertise well-established in the knowledge of artisanal rules does not mean that there is no room for diversity of individuality of craftsmanship’s virtuosity. This is the key idea behind this research. The differences in skill and virtuosity must be seen in the individual motor skill abilities of the creator-craftsman as well as in the cultural influence that the guild's models of learning have on the craftsman and that is produced in the novice-tutor relationship.

What is interesting about this study is that, taking into account the Aristotelian division, it seems that, unlike art understood as mimesis, where originality and novelty play an important role in the creative process (e.g. modern painting, poetry), a virtuoso craftsmanship (including pottery), i.e. one that has a higher aesthetic value (assuming the universalistic assumption that "beauty is not in the eye of the beholder", Dutton 2002), is created when cultural constraints are stronger (when tutor insist their novices accurately reproduce their gestures). I think that the future research indicated in the Conclusion could verify this hypothesis and the results would be extremely important not only for archeology but for both crosscultural aesthetics and anthropological study of art.

The research also shows that while the adherence of craftsmen to certain basic motor behaviors probably has an evolutionary foundation, variations within repetitive patterns are certainly influenced by individual diversity of motor skills. On the contrary, cultural influences are depended on models of learning in novice-expert personal relationship.

All these conclusions go far beyond the framework of archaeology and can be helpful for representatives of other disciplines, what makes this study even more fascinating.

7. PLOS authors have the option to publish the peer review history of their article (what does this mean?). If published, this will include your full peer review and any attached files.

Reviewer #1: **Yes: **C. Jeffra

Reviewer #3: No

---

## [Editor Report · Acceptance letter]

22 Sep 2020

PONE-D-20-08524R1 

Assessing the influence of culture on craft skills: A quantitative study with expert Nepalese potters 

Dear Dr. Gandon:

I'm pleased to inform you that your manuscript has been deemed suitable for publication in PLOS ONE. Congratulations! Your manuscript is now with our production department. 

Kind regards, 

on behalf of

Dr. Piotr Sorokowski 

Academic Editor

PLOS ONE